# Where Biology Meets Engineering: Scaling Up Microbial Nutraceuticals to Bridge Nutrition, Therapeutics, and Global Impact

**DOI:** 10.3390/microorganisms13030566

**Published:** 2025-03-02

**Authors:** Ahmed M. Elazzazy, Mohammed N. Baeshen, Khalid M. Alasmi, Shatha I. Alqurashi, Said E. Desouky, Sadat M. R. Khattab

**Affiliations:** 1Department of Biological Science, College of Science, University of Jeddah, Jeddah 21589, Saudi Arabia; mnbaeshen@uj.edu.sa (M.N.B.); kalasmi0007.stu@uj.edu.sa (K.M.A.); saaqurshi@uj.edu.sa (S.I.A.); 2Laboratory of Microbial Technology, Division of Systems Bioengineering, Department of Bioscience and Biotechnology, Faculty of Agriculture, Graduate School, Kyushu University, 744 Motooka, Nishi-ku, Fukuoka 819-0395, Japan; 3Institute of Advanced Energy, Kyoto University, Gokasho, Uji 611-0011, Japan; 4Research Institute for Sustainable Humanosphere, Kyoto University, Gokasho, Uji 611-0011, Japan

**Keywords:** microbial biotechnology, nutraceuticals, sustainable production, probiotics, prebiotics, postbiotics

## Abstract

The global nutraceutical industry is experiencing a paradigm shift, driven by an increasing demand for functional foods and dietary supplements that address malnutrition and chronic diseases such as obesity, diabetes, cardiovascular conditions, and cancer. Traditional plant- and animal-derived nutraceuticals face limitations in scalability, cost, and environmental impact, paving the way for microbial biotechnology as a sustainable alternative. Microbial cells act as bio-factories, converting nutrients like glucose and amino acids into valuable nutraceutical products such as polyunsaturated fatty acids (PUFAs), peptides, and other bioactive compounds. By harnessing their natural metabolic capabilities, microorganisms efficiently synthesize these bioactive compounds, making microbial production a sustainable and effective approach for nutraceutical development. This review explores the transformative role of microbial platforms in the production of nutraceuticals, emphasizing advanced fermentation techniques, synthetic biology, and metabolic engineering. It addresses the challenges of optimizing microbial strains, ensuring product quality, and scaling production while navigating regulatory frameworks. Furthermore, the review highlights cutting-edge technologies such as CRISPR/Cas9 for genome editing, adaptive evolution for strain enhancement, and bioreactor innovations to enhance yield and efficiency. With a focus on sustainability and precision, microbial production is positioned as a game-changer in the nutraceutical industry, offering eco-friendly and scalable solutions to meet global health needs. The integration of omics technologies and the exploration of novel microbial sources hold the potential to revolutionize this field, aligning with the growing consumer demand for innovative and functional bioactive products.


**Introduction**


The evolving global health landscape is characterized by a significant rise in diseases associated with malnutrition, largely driven by the increasing consumption of junk food. Conditions such as obesity, diabetes, cardiovascular disorders, and cancer have spurred a growing consumer interest in functional foods and nutraceutical products designed not only to nourish but also to prevent or mitigate disease [1,2].

Nutraceuticals, derived from food sources, offer health benefits beyond basic nutrition and encompass therapeutic properties such as antioxidant, anti-inflammatory, and antimicrobial activities [3]. Coined by Stephen DeFelice in 1989, the term “nutraceutical” merges “nutrition” and “pharmaceutical”, reflecting its dual role in health and therapy. These products are often described as dietary supplements delivering concentrated bioactive agents in non-food matrices at dosages exceeding those obtainable from regular food [4]. Unlike whole foods, nutraceuticals focus on isolated bioactive components, blurring the line between nutrition and pharmacology.

A more refined definition by [5] emphasizes nutraceuticals as substances cultivated, extracted, or synthesized under controlled conditions that, when administered orally, restore altered body structures and functions, improving overall health and well-being. This potential to prevent and treat chronic diseases, including cardiovascular diseases, cancer, and neurodegenerative disorders, underscores the growing global interest in nutraceuticals.

Traditionally, plant and animal sources have been the primary means of obtaining nutraceuticals. However, these sources face challenges such as seasonal variability, high extraction costs, and limited scalability [6]. In contrast, microbial production offers year-round consistency, easier genetic manipulation, and lower production costs, making it an attractive alternative. Microorganisms have emerged as critical platforms for producing bioactive compounds such as probiotics, peptides, polyunsaturated fatty acids (PUFAs), and polyphenols [5].

Microbial-based nutraceuticals hold immense promises for addressing the limitations of plant- and animal-based production. These techniques rely on fermentation technologies, where microorganisms either directly synthesize bioactive compounds or convert substrates into value-added products. This approach simplifies production, reduces costs, and enhances scalability. Microbial cells also serve as ideal hosts for genetic engineering, enabling the optimization of fermentation processes (Figure 1) and the utilization of simple carbon sources for efficient metabolite production [7]. Microbial nutraceuticals encompass a wide range of bioactive compounds, including vitamins, oligosaccharides, peptides, and pigments, many of which are already being commercially produced. These compounds offer sustainable and efficient alternatives to traditional sources and align with the global push for eco-friendly production practices.

This review will explore the current technologies, challenges, and future trends in microbial nutraceutical production. By focusing on these bio-based solutions, it aims to highlight how fermentation-driven microbial platforms are transforming the nutraceutical industry, ensuring sustainability and scalability while addressing global health challenges.


**Microbial production of nutraceuticals**


## 1. Probiotics

Probiotics are live microorganisms that, when consumed in adequate amounts, provide health benefits, particularly by improving gut health and supporting the immune system [8,9]. The term “probiotic”, derived from the Greek meaning “for life”, was first coined by Lilly and Stilwell in contrast to “antibiotic” [10]. Probiotics, including lactic acid bacteria (LAB) and bifidobacteria, are commonly used as food additives in dairy products and fruit juices to balance gut microbiota and treat gastrointestinal conditions such as Crohn’s disease [11,12]. Probiotics compete with harmful microorganisms for nutrients and adhesion sites, thereby preventing the colonization of pathogenic bacteria in the gut. Strains like Lactobacillus and Bifidobacterium encourage the production of antimicrobial agents such as bacteriocins, which inhibit harmful bacteria like *Clostridium difficile* and *Escherichia coli* [13]. Additionally, probiotics enhance the intestinal epithelial barrier by promoting the production of mucus and tight junction proteins, including occludins and claudins. This helps reduce intestinal permeability, commonly known as ’leaky gut’, and prevents the movement of pathogens and toxins from the gut into the bloodstream. For instance, *Lactobacillus plantarum* has been shown to increase the expression of tight junction proteins, thereby improving gut barrier function [14]. Probiotic bacteria aid the host’s nutrition by producing essential vitamins, particularly B vitamins such as folate (B9) and riboflavin (B2), which are crucial for energy metabolism and DNA synthesis. Additionally, certain probiotics possess antioxidant capabilities, helping to neutralize reactive oxygen species, thereby reducing oxidative stress and protecting cells from damage [15].

Strain specificity is a critical aspect of probiotics, as different strains of the same species can have unique health effects. *Lactobacillus rhamnosus* GG has been widely studied and is an effective approach to reducing the risk of upper respiratory tract infections in children attending daycare centers [16]. In contrast, *Lactobacillus plantarum* may be more beneficial for enhancing gut barrier function and reducing symptoms of irritable bowel syndrome due to its ability to modulate immune responses and strengthen tight junctions between intestinal cells [17]. Similarly, *Bifidobacterium longum* is recognized for its anti-inflammatory properties and its role in reducing symptoms of inflammatory bowel diseases like ulcerative colitis and Crohn’s disease. This strain can modulate the immune system and reduce intestinal inflammation, offering therapeutic benefits for individuals with gut disorders [18]. These examples underscore the importance of selecting the right strain for specific health conditions. Probiotic effectiveness is highly dependent on the strain, and not all probiotics will have the same therapeutic impact across different medical issues. Therefore, understanding strain-specific benefits is crucial for targeted probiotic therapy and achieving optimal health outcomes [19].

The global nutraceutical market grew from $274 billion to $292 billion in 2021, with 42% of consumers purchasing more functional foods compared to 2020 [20]. This increasing demand is driving the need for new technologies that allow for high-yield production of probiotics on a large scale [21]. While traditional batch and fed-batch fermentation processes are commonly used, continuous fermentation has shown advantages in producing *Bifidobacterium longum*, offering higher cell yields and reduced downstream processing [22]. Modified continuous or fed-batch fermentation with cell recycling can improve cell density, while techniques like DO-stat and exponential feeding can enhance biomass production [23,24].

## 2. Prebiotics and Postbiotics

The treatment of diseases and the enhancement of human health have historically relied on medicines derived from natural sources, synthetic compounds, and biologics, with advancements in chemistry, biology, and pharmacology transforming healthcare over the past two centuries. However, the interaction between medicines and gut microbiota has revealed broader physiological effects, challenging the traditional view of drug specificity [19,25]. These findings underscore the limitations of traditional medicines in addressing diseases linked to gut microbiota dysregulation and highlight the need for innovative therapeutic approaches [25,26]. Prebiotics and postbiotics have emerged as promising nutraceutical solutions to regulate gut microbiota and maintain intestinal health [27,28]. Prebiotics, non-digestible dietary components that selectively nourish beneficial bacteria, include compounds like inulin, fructooligosaccharides, galactooligosaccharides, and lactulose, found in foods such as garlic, onions, and bananas, and increasingly sourced from unconventional materials like seaweeds and microalgae [26,27]. Postbiotics, defined by the International Scientific Association for Probiotics and Prebiotics as preparations of inanimate microorganisms or their bioactive components, include short-chain fatty acids, peptides, and bacteriocins [19,26]. Unlike probiotics, postbiotics do not rely on live microorganisms, making them more stable and easier to incorporate into health interventions. Together, prebiotics and postbiotics work synergistically with probiotics, either promoting their growth or being derived from their metabolic activity, collectively enhancing gut microbiota balance and addressing diseases [26]. The growing market for probiotics, prebiotics, and postbiotics is driven by increasing consumer awareness of gut health and advances in microbiome research, with applications in metabolic disorders, mental health, and immune regulation. Future developments in production efficiency, fermentation techniques, and targeted delivery systems are poised to further expand their potential, ensuring their integration into mainstream healthcare and enhancing their role in addressing global health challenges [25,26]. The vital role of probiotics in promoting optimal health is presented in Figure 2, illustrating their beneficial effects.

## 3. Polyunsaturated Fatty Acids

Polyunsaturated fatty acids (PUFAs) are essential components of cell membranes and play a vital role in maintaining brain, nerve, and eye health. They also help prevent cardiovascular diseases, neurodegenerative disorders, and inflammatory diseases. Since the human body cannot synthesize omega-3 and omega-6 PUFAs, they must be obtained through diet [29,30].

PUFAs exhibit significant therapeutic potential, making them integral in the field of nutraceuticals. For example, eicosapentaenoic acid (EPA) demonstrates antimicrobial and antioxidant properties and improves lipid metabolism, reducing hepatic steatosis in non-alcoholic fatty liver disease [31]. Docosahexaenoic acid (DHA), which constitutes over 90% of brain omega-3 PUFAs and about 20% of total brain lipids, is critical for brain development and neuroprotection. DHA is incorporated into phosphatidylcholine and phosphatidylserine, influencing neurotransmitter release, neuronal growth, and gene expression [29,30]. Arachidonic acid (AA), a widely distributed PUFA, plays a crucial role in maintaining the fluidity of mammalian cell membranes. It is primarily derived from linoleic acid and serves as a precursor for numerous bioactive metabolites, including prostaglandins (PGs), thromboxanes (TXs), lipoxins (LXs), hydroxyeicosatetraenoic acids (HETEs), leukotrienes (LTs), and epoxyeicosatrienoic acids (EETs), through distinct metabolic pathways. In light of a rapidly aging global population, recent studies have highlighted the critical role of AA metabolism in the pathophysiology of aging-related diseases, such as osteoporosis and chronic obstructive pulmonary disease, among others [32].

A deficiency in essential fatty acids has been clearly linked to various health issues, including dermatitis, renal hypertension, mitochondrial dysfunction, cardiovascular diseases, type II diabetes, impaired brain development, arthritis, depression, and reduced immune resistance to infections [29]. Given this context, the health benefits of PUFAs, coupled with growing consumer awareness, are driving rapid growth in the global PUFA nutraceutical market, which was valued at US $6019 million in 2023 and is expected to reach US $9717.8 million by 2033, with a compound annual growth rate CAGR of 5.4% [33]. The metabolic pathway for a PUFA is depicted in Figure 3.

Despite increasing demand, production bottlenecks remain a challenge. For instance, global industrial capacity for AA production is expected to reach 410 thousand tons by 2025, yet this will still fall short of market needs [33]. Similarly, there is an anticipated deficit of over one million tons of omega-3 PUFAs, and DHA shortages may deprive nearly 90% of the global population of this critical nutrient [34].

Traditional sources of omega-3 PUFAs, such as fish oil, face sustainability concerns due to overfishing. As a solution, microbial production of PUFAs using microalgae and oleaginous microorganisms offers a more sustainable, environmentally friendly alternative. Microalgae species like *Schizochytrium* sp. and *Aurantiochytrium* sp. naturally accumulate high levels of long-chain PUFAs (LC-PUFAs), particularly DHA, with metabolic engineering leading to 40–100% increases in DHA yields [33]. Other microalgae, including *Isochrysis*, *Nannochloropsis*, and *Phaeodactylum*, have emerged as viable producers of omega-3 LC-PUFAs [35,36]. Microorganisms like *Crypthecodinium cohnii* and *Schizochytrium* sp. have achieved high DHA yields up to 13.3 g/L [33].

Oleaginous microorganisms, including *Yarrowia lipolytica*, *Mortierella alpina*, and various microalgae, are favored for large-scale PUFA production due to their ability to accumulate more than 20% lipids in their biomass [37]. For instance, *Mortierella alpina* is known for its high production of arachidonic acid (ARA), an omega-6 PUFA with various applications in medicine, cosmetics, and food [38]. Table 1 provides a summary of literature reports on oleaginous organisms and their respective oil accumulation capabilities.

However, challenges remain in optimizing the large-scale production of microbial PUFAs, including cost-effective fermentation processes, efficient extraction methods, and yield optimization. Research continues to explore the use of low-cost carbon sources, such as agricultural residues and food waste, to improve economic viability [7]. To address these challenges and enhance PUFA production, advances in bioreactor design, microbial strain development, extraction techniques, and metabolic engineering have been employed to optimize PUFA synthesis in various organisms. Microalgae, such as *Schizochytrium* sp. and *Aurantiochytrium* sp., have emerged as promising candidates due to their natural ability to accumulate high levels of long-chain PUFAs (LC-PUFAs). Notably, metabolic engineering efforts have led to 40–100% increases in DHA yields in these species [33]. *Yarrowia lipolytica*, *Mucor circinelloides* and *Mortierella alpina* has been genetically engineered to achieve a high lipid content reaching up to 30% of the dry cell weight (DCW) [49]. *S. cerevisiae* engineered for PUFA production by introducing desaturase and elongase genes from microalgae and fungi, enabling them to synthesize omega-3 PUFAs such as ALA, EPA, and DHA.

## 4. Exopolysaccharides (EPSs)

Exopolysaccharides (EPSs) are bioactive polymers that play a vital role in human health, exhibiting prebiotic, anticancer, anti-ulcer, immunomodulatory, cholesterol-lowering, antioxidant, antimicrobial, antibiofilm, and antihypertensive properties. Beyond their bioactivity, EPSs demonstrate immense potential in therapeutic delivery systems, serving as carriers for drugs, probiotics, and other bioactive agents [50,51]. EPSs exert their bioactivity through interconnected pathways: they selectively modulate gut microbiota by stimulating beneficial bacterial strains and enhancing short-chain fatty acid (SCFA) production, mitigating oxidative stress through direct ROS scavenging and activation of the Nrf2/ARE signaling cascade, and fine-tuning immune responses by engaging Toll-like receptors (TLRs), which initiate cytokine secretion, macrophage polarization, and adaptive immune modulation [50,51]. Their unique physicochemical properties, such as adhesion, water-binding capacity, and hydrogel-forming ability, enable them to act as protective and controlled-release systems, while their biodegradability and safety further enhance their applications. Notable examples of EPS utilization include their role in vaccine preparations as antigen carriers or adjuvant systems, in gene therapy as gene delivery vectors, and in encapsulation systems to improve cell stability and viability, as evidenced by the successful use of alginate beads for encapsulating fibroblast cells and probiotic bacterial strains [52]. Microbial polysaccharides such as xanthan, gellan, dextran, and alginate have been successfully commercialized due to their wide-ranging industrial applications in food, pharmaceuticals, and cosmetics. Xanthan gum, produced by *Xanthomonas campestris*, is extensively used as a thickening agent and stabilizer in food processing, while gellan gum from *Sphingomonas elodea* finds use in beverages and as a gelling agent in cosmetics and personal care products [53].

Apart from bacterial polysaccharides, fungi also play a key role in EPS production. For example, the fungus *Sclerotium rolfsii* produces scleroglucan, a unique extracellular polysaccharide with demonstrated antitumor and antiviral activities, making it a promising candidate for therapeutic applications [53]. Microbial sources such as *Sclerotium rolfsii* provide advantages like scalability, cost-effective production, and lower environmental impact compared to plant- or animal-derived polysaccharides. To enhance scleroglucan production, various bioprocess optimization strategies have been explored. For instance, a pH-shift strategy significantly improved scleroglucan production from 32.4 g/L to 42 g/L, with a productivity rate of 0.5 g/L/h [54]. Further improvements were achieved using optimized fed-batch fermentation strategies, leading to an impressive scleroglucan yield of 66.6 g/L [55]. These advancements underscore the feasibility of large-scale microbial EPS production, reducing dependency on traditional sources and providing a sustainable alternative for diverse industries, including pharmaceuticals and cosmetics.

Additionally, animal-derived polysaccharides such as chondroitin sulfate, hyaluronic acid (HA), and heparosan have gained attention for their biomedical applications, including tissue engineering and regenerative medicine. These polysaccharides are now being biosynthesized in host organisms like *E. coli*, *Lactococcus lactis*, and *Streptomyces albulus* through metabolic engineering and fermentation technologies [56]. For instance, *L. lactis* strains have been engineered to produce HA, a polysaccharide critical for skin hydration and joint health. By co-expressing key enzymes such as glucose pyrophosphorylase and glucose dehydrogenase, HA production levels were significantly increased to 1.8 g/L, representing a breakthrough in the sustainable microbial production of HA [5,56].

### 4.1. Microbial EPS Biosynthesis and Metabolic Pathways

Microbial EPS biosynthesis varies across species and depends on specific metabolic pathways and substrate utilization. The sucrose polymerization pathway, utilized by *Lactobacillus* and *Leuconostoc* species, involves dextransucrase and levansucrase, which directly convert sucrose into dextran and levan, respectively, without requiring nucleotide sugar intermediates. This mechanism enhances efficiency and yield in microbial EPS production. Another significant route, the nucleotide–sugar pathway, is utilized in the biosynthesis of xanthan, alginate, and hyaluronic acid. Key precursors such as UDP-glucose, UDP-galactose, and GDP-mannose form EPS monomers through the coordinated action of phosphoglucomutase, UDP-glucose pyrophosphorylase, and glycosyltransferases [51].

In Gram-negative bacteria, the Wzx/Wzy-dependent polymerization pathway plays a crucial role in EPS synthesis. Here, monomers are synthesized intracellularly, transported via Wzx flippase, and polymerized by Wzy polymerase, enabling regulated production of high-molecular-weight EPSs [51,52]. EPS biosynthesis is further modulated by global transcriptional regulators, such as RpoS in Gram-negative bacteria and Spo0A in *Bacillus* species, which respond to environmental and metabolic cues to optimize EPS production [51]. See Refs. [50,51,52] for further details.

### 4.2. Industrial Applications and Commercial Value

Microbial EPSs exhibit remarkable versatility, extending beyond their bioactive functions to diverse applications in pharmaceuticals, nutraceuticals, food technology, and biomaterials [50,51]. Hyaluronic acid (HA), widely used in wound healing and cosmetic formulations, has successfully transitioned from animal-derived to microbial production, achieving higher purity and efficiency [56]. Similarly, chondroitin sulfate and heparosan, previously obtained from cartilage and animal tissues, are now microbially synthesized, ensuring sustainable production with consistent quality [57]. Table 2 presents a comparative analysis of microbial EPSs, detailing their biosynthetic origins, enzymatic pathways, bioactive mechanisms, and industrial significance. The inclusion of antioxidant, immunomodulatory, and prebiotic properties further illustrates the multifunctionality of EPSs and their growing applications in biomedicine, functional foods, and sustainable biomaterials.

### 4.3. Future Perspectives and Next-Generation EPS Production

Despite significant progress, key challenges, including high production costs, purification bottlenecks, and regulatory complexities continue to hinder large-scale microbial EPS commercialization. Future research should prioritize cost-effective substrates, high-throughput fermentation, and process intensification to improve yield, efficiency, and commercial scalability. Furthermore, streamlining regulatory frameworks for microbial EPSs in food, pharmaceuticals, and biomaterials will be crucial for accelerating global adoption and market integration [50,52]. Advances in precision fermentation, adaptive laboratory evolution, and bioprocess intensification strategies are paving the way for high-yield EPS production with a reduced environmental footprint. These innovations, coupled with metabolic modeling and in silico pathway optimization, will enable more efficient microbial strains tailored for industrial applications. CRISPR-based metabolic engineering has transformed polysaccharide biosynthesis by enabling targeted gene modifications that optimize metabolic flux, enhance yield, and increase microbial robustness for industrial applications [58]. Concurrently, cell-free enzymatic systems represent a paradigm shift in EPS production, facilitating one-pot enzymatic polymerization, reducing energy-intensive purification steps, and streamlining large-scale bioprocessing for next-generation EPS applications [58]. With continuous improvements in metabolic engineering, process optimization, and synthetic biology, microbial EPS production is emerging as a viable and sustainable alternative to traditional sources [50]. These advances are expected to drive industrial adoption and long-term commercial success, ensuring that microbial EPSs remain at the forefront of biomedicine, food technology, and sustainable materials [50,51,58].

## 5. Oligosaccharides

Low-molecular-weight, non-digestible carbohydrates known as oligosaccharides possess prebiotic, antibiotic, anticancer, antidiabetic, antihypertensive, and anti-inflammatory properties, which are beneficial for health. Naturally occurring in milk, cereals, vegetables, and honey, oligosaccharides are also produced industrially through chemical and microbial processes. Microbial methods, utilizing whole cells and microbial enzymes, have advantages over chemical processes, including economic viability and environmental sustainability [59].

Currently, more than 13 different types of functional oligosaccharides are commercially produced, with fructooligosaccharides (FOS), galactooligosaccharides (GOS), and xylooligosaccharides (XOS) being the most prevalent [59]. Research using *Aspergillus ibericus* to optimize fermentation conditions in a two-step process for FOS production has shown improvements in yield and efficiency [60]. Additionally, *Bacillus subtilis* has been effectively utilized to produce high-purity XOS from sugarcane bagasse through the action of xylanase. These examples highlight the diversity of microorganisms and enzymes employed in the industrial production of oligosaccharides. Functional oligosaccharides are frequently produced using microbial enzymes. Invertase from *Saccharomyces cerevisiae* is commonly used for FOS production, while other enzymes facilitate the conversion of various substrates into functional oligosaccharides. The ability of whole microbial cells and their enzymes to produce high-quality oligosaccharides adds to the efficiency of these processes [37].

Despite these advances, challenges remain in the large-scale production of oligosaccharides. Issues such as yield optimization, cost reduction, and regulatory considerations must be addressed to enhance the commercial viability of these products. Continuous improvements in fermentation techniques, including fed-batch and continuous fermentation, as well as advances in bioreactor technologies, are essential for maximizing production efficiency. Additionally, expanding the scope of microbial sources could yield new insights and applications [61]. Exploring other fungi and bacteria that produce oligosaccharides, or their enzymes, can lead to novel products with enhanced functional properties [37]. Research into the specific health benefits of oligosaccharides, such as the prebiotic effects of GOS and FOS on gut health, can further validate their significance in health and nutrition [59]. In summary, the microbial production of oligosaccharides represents a promising avenue for sustainable and functional carbohydrate sources, with ongoing research likely to unlock new applications and improve production methodologies [61].

## 6. Low-Calorie Sugars

Obesity has been projected to contribute 2–5% of total healthcare expenditures, increasing the demand for food products designed to lower calorie consumption. Low-calorie sweeteners, such as polyols, are gaining popularity as sugar substitutes because they provide a similar sweetness profile while containing significantly fewer calories. Common polyols like mannitol and sorbitol mimic the taste and sweetness of traditional sugars such as sucrose and glucose [62]. Beyond their reduced caloric content, mannitol also functions as an antioxidant, enhancing the viability of biological cells during freezing or drying processes [63,64]. Certain LAB, such as *Leuconostoc mesenteroides*, are well-known for their ability to produce mannitol during fructose fermentation. These bacteria directly convert most of the fructose into mannitol, while the remainder is utilized for energy production [61,64]. Such advantages of utilizing *L. mesenteroides* were applied for improving the quality of kimchi production [64]. Yeasts such as *Candida magnoliae* and *Candida parapsilosis* also showed efficient capabilities to produce mannitol from glucose [65].

Polyols such as sorbitol, erythritol, xylitol, isomalt, and lactitol have garnered increasing attention for their health-promoting properties and broad applications across industries. Sorbitol offers two-thirds of the caloric content of sucrose with 60% of its sweetness, making it a popular alternative. *Lactobacillus plantarum* has been reported to produce sorbitol with a 65% conversion yield. Moreover, a recombinant *Zymomonas mobilis* strain has demonstrated the ability to convert fructose completely (100%) into sorbitol while simultaneously converting glucose into gluconic acid [66]. Additionally, the native *Z. mobilis* strain was able to produce 31.2 g/L and 30.4 g/L of sorbitol from sugarcane molasses and sugarcane bagasse, respectively [67]. Erythritol has moderate sweetness (60–80% that of sucrose) and is almost calorie-free and does not affect blood glucose levels, making it suitable for people with diabetes [67]. Erythritol, a widely recognized functional sweetener, was first identified in 1848. Japan authorized its use as a sweetener in 1990, and by the early 21st century, it had been approved in the United States and various European countries. In 2008, China adopted standards for erythritol, indicating its vast potential for broad applications [68]. Several microbial cell factories exhibited different capabilities to produce erythritol from glucose or glycol, such as *Candida magnoliae*, *Aureobasidium* sp., *Torula* sp., *Moniliella* sp., *Pseudozyma tsukubaensis*, and *Yarrowia lipolytica* [62]. Their production was varied according to strain as listed in Table 2. Owing to strong capabilities of strains, erythritol was produced industrially via some of these strains, especially *Y. lipolytica*, whose capability was further improved via metabolic engineering [49,68]. The metabolic pathway for erythritol production is illustrated in Figure 4.

The development and commercial use of these low-calorie polyols have become critical in creating food products aimed at combating obesity and related health issues. Their widespread use in food and beverage industries offers health-conscious consumers alternatives that help reduce calorie intake while maintaining the sensory qualities of traditional sugary products [61]. By incorporating low-calorie sweeteners into everyday food items, industries are addressing both the growing obesity epidemic and the demand for healthier, functional food products [62].

## 7. Lactic Acid

Lactic acid is a versatile compound with applications spanning diverse sectors, including pharmaceuticals, cosmetics, food industries, specialty chemicals, textiles, and leather production [69]. The global demand for lactic acid (LA) is anticipated to rise significantly, increasing from approximately 1220 kilotons in 2016 to 1960 kilotons by 2025 [70]. This substantial growth underscores the economic importance of LA, with the market value expected to reach around $5.8–6.7 billion by 2030, reflecting its expanding industrial applications. Enhancing biotechnological processes for LA production has become a priority to improve both energy efficiency and sustainability [71].

Fermentation technology and metabolic engineering have advanced significantly, and over 90% of LA is now produced via fermentation. Major producers include Corbion and Cargill, which produce 240,000 and 180,000 tons, respectively [72]. Lactic acid bacteria (LAB), particularly *Lactococcus lactis*, are commonly used for fermentation, contributing flavor, texture, and nutrition, especially in dairy products [73]. Key factors in LA production include pH control, sugar concentration, and nutrient availability. Downstream processing, which involves purification and recovery, can contribute up to 50% of LA’s final cost due to complex neutralization and waste management [74].

## 8. Polyphenols

Polyphenols are naturally occurring, plant-derived bioactive compounds that have attracted considerable interest in recent years due to their health-promoting properties. These compounds, characterized by their structural diversity, are predominantly found in fruits, vegetables, and other plant-based foods. Polyphenols play a crucial role in preventing and alleviating symptoms of various diseases, offering benefits such as antioxidant, anti-inflammatory, and antimicrobial activities [75]. However, a significant limitation of polyphenols is their low bioavailability, which reduces their therapeutic efficacy despite their numerous beneficial effects [76].

Microbial biotransformation has emerged as a promising approach to enhance the bioavailability of polyphenols. Using microorganisms such as *Lactobacillus plantarum*, *Lactobacillus brevis*, *Bacillus* spp., *Saccharomyces cerevisiae*, and *Aspergillus niger*, polyphenols can be converted into metabolites with improved solubility, stability, and bioactivity [77]. Polyphenols, as prominent bioactive compounds in edible fungi such as *Lentinula edodes* (shiitake), *Pleurotus ostreatus* (oyster mushroom), *Agaricus bisporus* (white button mushroom), and *Ganoderma lucidum* (reishi mushroom), have garnered attention for their significant nutritional and medicinal value. Notable fungal-derived polyphenols, such as quercetin and kaempferol, exhibit potent antioxidant, anti-inflammatory, and anticarcinogenic properties [12]. These health-promoting effects are attributed to their unique structures and mechanisms of action, which enhance their therapeutic potential. Additionally, the safety profile of polyphenols derived from dietary fungi supports their inclusion in dietary interventions and therapeutic applications. Mushroom-based polyphenols are not only valued for their consumption as part of a regular diet but are increasingly utilized in the nutraceutical and pharmaceutical industries [78,79].

## 9. Selenium

Selenium exists in two main forms: organic and inorganic. Organic selenium compounds include seleno-methionine (SeMet) and seleno-cysteine, which are present in selenium-rich foods such as meat, fish, eggs, and nuts. SeMet is incorporated into proteins in place of methionine and plays a key role in the body’s antioxidant systems [54]. Microorganisms accumulate selenium through extracellular binding on cell walls and membranes and intracellular transport across biological membranes. Selenium-enriched yeast, particularly *Saccharomyces cerevisiae*, is widely used due to its low toxicity and high bioavailability, making it an ideal selenium supplement in the feed industry. It promotes animal health, enhances immune responses, and improves the quality of animal products, meeting the demand for biofortified supplements [80].

Advances in fermentation technology have further improved the productivity of selenium-enriched yeast. By optimizing factors such as pH, temperature, and nutrient levels, Se-yeast biomass production has been increased, making it suitable for inclusion not only in supplements but also in functional foods such as bread and cereals [81]. These advances have expanded the applications of selenium-enriched yeast as a safe and effective way to meet selenium dietary requirements while reducing the risks associated with toxicity.

## 10. Vitamins

Microbial production of vitamins has emerged as a superior alternative to traditional chemical synthesis due to its flexibility in microorganism selection, the use of renewable biomass, and environmentally sustainable processes. Various microorganisms, including *Bacillus subtilis*, *Propionibacterium freudenreichii*, *Saccharomyces cerevisiae*, and *Rhizopus oryzae*, are widely utilized for the biosynthesis of essential vitamins such as riboflavin (B2), cobalamin (B12), folate (B9), and biotin (B7) [82,83]. These microbial processes offer high yield, cost-effectiveness, and improved bioavailability, making them a preferred method for vitamin production in the nutraceutical and pharmaceutical industries.

### 10.1. Riboflavin (Vitamin B12)

Riboflavin is an essential water-soluble vitamin that plays a critical role in energy production and cellular metabolism. Microbial production of riboflavin is now considered a sustainable and eco-friendly alternative to chemical synthesis, with *Bacillus subtilis* and *Ashbya gossypii* being the most prominent microbial producers. Other microorganisms like *Streptomyces*, *Methanobacterium*, and *Klebsiella* are now employed to meet the rising global demand for vitamin B12. *Propionibacteria* is particularly preferred for its safety and efficiency in production. Advances in genetic engineering—such as precursor addition and trace metal supplementation—have significantly enhanced large-scale vitamin B12 production, especially through microbes like *Bacillus subtilis* and *Ashbya gossypii* [84]. Riboflavin has also been successfully produced from renewable carbon sources like agricultural waste. Genetic modifications have increased riboflavin yields by up to 5.4-fold in *A. gossypii*, while *Candida famata* has produced up to 16.4 g/L of riboflavin under optimized conditions. Additionally, lactic acid bacteria (LAB), such as *Lactobacillus reuteri*, have shown promise in producing vitamin B12, offering potential for fortified fermented food products [85].

### 10.2. Vitamin C (Ascorbic Acid)

Traditionally, vitamin C was chemically synthesized, but microbial production is gaining traction due to its sustainability. A two-step fermentation process involving *Gluconobacter oxydans* and *Ketogulonigenium vulgare* has been developed for large-scale production of ascorbic acid. The process starts with *G. oxydans* converting glucose to 2-keto-L-gulonic acid (2-KLG), which is then transformed into vitamin C by *K. vulgare*. Studies have enhanced this process by using metabolic engineering to increase yields and reduce costs [86]. Direct production of vitamin C was also explored by engineering the metabolic pathways of *Saccharomyces cerevisiae*, achieving a production yield of 44 mg/L [87].

### 10.3. Coenzyme Q10 (Ubiquinone)

Coenzyme Q10 consists of a quinone moiety linked to a tail made up of ten isoprene units and is known for its antioxidant properties. Clinical studies have indicated that a deficiency in CoQ10 may lead to several health conditions, including heart disease, cancer, dental issues, Parkinson’s disease, and migraine headaches [88]. This underscores the critical role of CoQ10 in overall health and its potential in preventing these disorders. Therefore, it is increasingly produced using microbial fermentation. *Rhodobacter sphaeroides* and *Agrobacterium tumefaciens* are widely employed for its production. Advances in strain improvement and fermentation optimization have increased yields. For instance, researchers have mutated strains or overexpressed genes in the CoQ10 biosynthesis pathway to enhance productivity, achieving yields of up to 300 mg/L [89,90].

### 10.4. Vitamin B7 (Biotin)

Biotin is a vital coenzyme, and each year, significant quantities are produced chemically for commercial applications. This biotin is utilized as a dietary supplement and as an essential ingredient in pharmaceuticals and cosmetics [91]. In recent years, microbial production of biotin has garnered attention as a more sustainable alternative [92]. Researchers have optimized the biosynthetic pathways for biotin in various microorganisms by manipulating regulatory networks and precursor availability. For instance, one study successfully engineered *Escherichia coli* to produce up to 208.7 mg/L of biotin through fed-batch fermentation, marking a notable advancement over traditional methods [93].

### 10.5. Carotenoids (e.g., β-Carotene)

Carotenoids are vital bioactive compounds with antioxidant, anti-inflammatory, and pro-vitamin-A properties, making them valuable for nutraceutical, pharmaceutical, food, and cosmetic industries. Although synthetic carotenoids dominate the market, concerns over chemical waste and health risks have increased interest in microbial production as a sustainable alternative. Microorganisms such as fungi (*Blakeslea trispora*, *Phycomyces blakesleeanus*), yeast (*Rhodotorula* spp., *Xanthophyllomyces dendrorhous*), bacteria (*Corynebacterium michiganense*, *Dietzia natronolimnaea*), and microalgae (*Haematococcus pluvialis*, *Dunaliella salina*) efficiently produce carotenoids under controlled conditions. *Rhodotorula* spp., widely studied for carotenoid biosynthesis, produces β-carotene, torulene, and torularhodin, with yields influenced by environmental conditions. Microbial fermentation offers advantages such as cost-effective production using low-cost substrates, season-independent processing, and high efficiency, positioning it as a promising alternative to synthetic carotenoids [82,83]. Yeast species like *Yarrowia lipolytica* have been engineered to overproduce carotenoids. Using metabolic engineering, researchers have increased carotenoid yields by overexpressing the genes involved in the mevalonate pathway [82,83].

### 10.6. Vitamin B9 (Folate)

Folate is known for its protective effects against inflammation and cancer, partly by addressing folate deficiency, which is linked to premalignant changes in the colonic epithelium [94]. Microbial folate production has gained attention for being eco-friendlier and more cost-effective than chemical synthesis. *Propionibacteria*, a Generally Recognized as Safe (GRAS) organism, produces more folate than *Streptococcus thermophilus* and is commonly used in starter cultures for folate-rich foods like cheese and milk [95]. Lactic acid bacteria (LAB), such as *Lactococcus lactis* and *Lactobacillus plantarum*, are efficient folate producers. Genetic engineering, including the use of a nisin-controlled expression system in *L. lactis*, has resulted in a threefold increase in folate production [95]. Modifying growth media, fermentation conditions, and genetic pathways has enabled researchers to boost folate production in LAB, facilitating the development of folate-enriched fermented dairy products like yogurt and cheese [96]. Probiotic bacteria from the *Lactobacillus* and *Bifidobacterium* genera are also studied for their ability to produce folate and promote health. Although many wild-type lactobacilli do not synthesize folate, *Lactobacillus plantarum* can produce folate in the presence of para-aminobenzoic acid. Moreover, bifidobacteria such as *B. adolescentis* and *B. pseudocatenulatum* have been shown to enhance folate levels in animal and human trials, supporting their use as probiotic supplements [97]. Folate has been shown to provide effective protection against inflammation and cancer by harnessing the beneficial properties of probiotics while also addressing folate deficiency, which is associated with premalignant changes in the colonic epithelium [97].

### 10.7. Vitamin E (Tocopherol)

Vitamin E (tocopherol) is a fat-soluble antioxidant that plays a key role in protecting cells from oxidative damage and supporting immune function. In *Synechocystis* sp. strain PCC 6803, tocopherol-deficient mutants were found to be more sensitive to lipid peroxidation under high-light stress, indicating that tocopherols play an essential role in protecting against lipid damage, especially when carotenoid synthesis is inhibited [98]. Vitamin E is commonly found in dietary supplements and skincare products. It is increasingly produced through microbial fermentation, an emerging field due to its potential cost-effectiveness and sustainability. *S. cerevisiae* was engineered using genes from photosynthetic organisms, optimized pathways, and a cold-shock-controlled two-stage fermentation, achieving yields of up to 320 mg/L, thus offering a promising approach for large-scale tocotrienol production [56]. Researchers have developed an innovative method for synthesizing vitamin E by converting microbial-fermented farnesene into isophytol using *Saccharomyces cerevisiae*. This process achieves a high yield of b-farnesene (55.4 g/L) and converts it into isophytol with a 92% efficiency, providing a cost-effective and safer alternative to traditional chemical synthesis. Since its implementation in 2017, this method has significantly transformed the vitamin E market, enabling an annual output of 30,000 tons in China [99]. The new vitamin E synthesis process using farnesene is competitive only if microbial fermentation costs are kept below $6/kg. Amyris has optimized farnesene production to achieve a titer of 130 g/L, with a glucose conversion rate of nearly 20%, bringing the carbon cost down to $2.3/kg [100]. Despite these advances, further reductions in fermentation costs can further improve the economic viability of this approach for vitamin E production [99].

### 10.8. Vitamin K

Vitamin K is a fat-soluble vitamin that exists in two primary forms: phylloquinone (PK) and menaquinone (MK). While PK is predominantly found in plant-based foods such as leafy green vegetables, MK, considered more bioactive, is primarily produced by microbes [101]. MK plays a crucial role in several physiological processes, including bone health and cardiovascular function, due to its involvement in calcium metabolism and coagulation.

*Bacillus subtilis* has been identified as a key strain for industrial MK production, particularly menaquinone-7 (MK-7), a form known for its high bioactivity and extended half-life in the body. Several studies have focused on optimizing fermentation processes to increase yields of MK-7, including efforts to enhance bacterial growth, substrate utilization, and vitamin production. For example, MK-7 production increased by 1.66 times in a mutated *B. subtilis* natto strain under optimized fermentation conditions, where factors like pH, aeration, and nutrient availability were carefully controlled [101]. Additionally, strains of *Lactococcus lactis*, a commonly used lactic acid bacterium in dairy fermentation, have shown the ability to produce long-chain MKs in various fermentation environments. Research has demonstrated the potential for these strains to be used in developing MK-enriched functional foods and supplements, potentially offering new avenues for fortifying foods with bioavailable vitamin K2 [102]. The rising demand for MK-enriched supplements and functional foods has driven interest in microbial production processes. The ability to harness microbes like *B. subtilis* and *Lactococcus lactis* for MK production offers a sustainable alternative to chemical synthesis or extraction from animal sources. Moreover, microbial fermentation is highly scalable and can be controlled to produce specific types of MK with high purity and bioactivity, making it a preferred method for industrial production.

## 11. Glutathione and α-Ketoglutarate

Microorganisms such as *S. cerevisiae*, *Candida utilis*, and *Yarrowia lipolytica* are vital players in the production of key nutraceuticals. *S. cerevisiae*, a GRAS organism, is particularly notable for its ability to produce glutathione (GSH), a powerful antioxidant composed of glycine, cysteine, and glutamate. Through techniques like random mutagenesis and high-cell-density fermentation, GSH production has been significantly enhanced. Recent studies have reported GSH yields reaching 75 mg/L, with a 55% increase in biomass by adding cysteine and glycine [103]. This highlights the industrial potential of yeast in producing GSH for nutraceutical and therapeutic applications.

Additionally, *Yarrowia lipolytica* has shown efficiency in producing α-ketoglutaric acid (α-KG), a key bioactive compound used in food, pharmaceuticals, and cosmetics. α-KG plays critical roles in energy metabolism and antioxidant protection. *Y. lipolytica* can produce α-ketoglutaric acid (KGA) using renewable carbon sources such as glycerol and rapeseed oil. By optimizing parameters like pH, aeration, and supplementation with thiamine, the process yielded 82.4 g/L of KGA with a productivity of 0.57 g/L/h, highlighting the potential of renewable substrates for efficient KGA biosynthesis with minimal byproducts [104].

## 12. Gamma-Aminobutyric Acid (GABA)

Gamma-aminobutyric acid (GABA) is a non-protein amino acid with inhibitory neurotransmitter functions, widely recognized for its diverse health benefits, including antihypertensive, immunomodulatory, antidiabetic, and neuroprotective effects [105,106]. GABA is increasingly incorporated into functional foods and nutraceuticals, such as yogurt and beverages, due to its ability to reduce anxiety, improve sleep, and manage hypertension. It also exhibits anti-inflammatory, antihypertensive, and antidiabetic effects, alongside potential neuroprotective and anticancer properties through mechanisms such as inducing cancer cell death and regulating tumor growth [105]. Furthermore, GABA intake may regulate the lateral hypothalamus, aiding in the management of obesity and overeating disorders, thus broadening its applications beyond neurological and cardiovascular health [107]. Despite its promise, challenges remain in optimizing fermentation processes to ensure safety and efficiency. Some LAB strains may produce harmful biogenic amines like histamine and tyramine during fermentation, posing potential health risks. Comprehensive studies, including in vitro and in vivo analyses, are essential to identify strains that are safe and yield high GABA levels without producing toxic byproducts [108]. Continued advancements in genetic engineering, fermentation technology, and regulatory standardization are critical for scaling GABA production and expanding its applications in the food and health industries. These efforts ensure the safe and effective integration of GABA into consumer products, aligning with its growing potential as a functional ingredient in addressing diverse health needs [105].

## 13. Microbial Pigments

Microbial pigments are valued for their chemical diversity, vibrant colors, and functional properties, offering a sustainable alternative to plant- and animal-derived pigments. These pigments, such as carotenoids, monascins, flavins, and prodigiosin, find applications in industries including food, cosmetics, textiles, nutraceuticals, and pharmaceuticals [109]. Beyond their role as natural colorants, microbial pigments exhibit health-promoting properties such as antioxidant, antimicrobial, and anticancer activities. For example, *Monascus* species produce red, orange, and yellow pigments in fermented rice products, while microbial carotenoids, important precursors to vitamin A, are favored for their higher yields, cost-effectiveness, and eco-friendly production processes [110]. The Microbial pigments in nutraceuticals and their health benefits is illustrated in Figure 5.

The global market for natural pigments is experiencing significant growth, with values projected to rise from $1.5 billion in 2020 to $2.5 billion by 2025, driven by high demand in Europe and the U.S. Meanwhile, the organic pigments market is expected to reach $8.4 billion by 2031, growing at a compound annual growth rate (CAGR) of 4.2% from 2022 to 2031 [111]. Certain microorganisms, such as *Micropeneta laotica* (dimorphic) and *Thermus* sp. (thermophilic), thrive in extreme environments, making them valuable sources of pigments. Strains like *Bacillus* sp., *Sarcina* sp., and *Streptomyces* sp. are recognized for producing pigments with bioactive properties, including those with roles as anticancer agents, mutagenesis inhibitors, and bio-indicators. Marine-derived pigments such as dolastatin, isolated from mollusks, also demonstrate potent antitumor activities [112]. Advances in metabolic pathway engineering, genetic engineering, and DNA sequencing are pivotal for optimizing pigment yields and functionality. Exploring unexplored niches and extreme environments may reveal novel microbial species with high-value pigment production capabilities. These innovations are essential for establishing microbial pigments as viable, sustainable alternatives to synthetic colorants across diverse industries [109].

## 14. Bioactive Peptides

Bioactive peptides (BAPs) are short amino acid sequences derived from proteins through proteolysis, exhibiting diverse biological activities, including antihypertensive, antimicrobial, antioxidant, antidiabetic, and immunomodulatory effects [113]. These peptides naturally occur in fermented foods such as milk, soybeans, fish, and cereals and are produced via microbial fermentation, enzymatic hydrolysis, or a combination of both. Their bioactivity and functionality are significantly influenced by microbial strain selection, substrate composition, and fermentation conditions [114].

### 14.1. Mechanisms of Bioactivity

BAPs exert their physiological effects through multiple molecular pathways, regulating key biological functions at the cellular and systemic levels.

#### 14.1.1. Antihypertensive Activity

The antihypertensive properties of BAPs are primarily mediated through angiotensin-converting enzyme (ACE) inhibition, which prevents the conversion of angiotensin I to angiotensin II, a vasoconstrictor that regulates blood pressure. Dairy-derived peptides, such as Val-Pro-Pro (VPP) and Ile-Pro-Pro (IPP), act as competitive ACE inhibitors, leading to vasodilation and reduced hypertension [115].

#### 14.1.2. Antioxidant Mechanism

BAPs exhibit antioxidant properties by scavenging reactive oxygen species (ROS) and activating the Nrf2/ARE signaling pathway, which regulates endogenous antioxidant enzymes, including superoxide dismutase (SOD) and glutathione peroxidase (GPx). This mechanism helps reduce oxidative stress and cellular damage [116].

#### 14.1.3. Immunomodulation

BAPs interact with Toll-like receptors (TLRs), particularly TLR2 and TLR4, which play a crucial role in innate immunity. These interactions trigger cytokine secretion and macrophage activation, modulating adaptive immune responses and inflammation control [117].

#### 14.1.4. Antimicrobial Activity

A specific class of BAPs, antimicrobial peptides (AMPs), disrupt bacterial membranes through electrostatic interactions, leading to cell lysis and inhibition of quorum sensing, a bacterial communication system crucial for biofilm formation and virulence factor expression [118]. Table 3 provides an overview of BAPs, summarizing their microbial sources, key bioactive mechanisms, and industrial applications.

### 14.2. Production and Industrial Applications

The microbial production of BAPs has been widely explored, particularly in protein-rich dairy and non-dairy foods. Microbial proteases selectively hydrolyze proteins into bioactive peptide fractions, enhancing their physiological potency.

The use of probiotic strains, such as *Lactobacillus* and *Bifidobacterium*, has been shown to increase bioactive peptide release during fermentation, improving their therapeutic potential. Certain peptides derived from *Lactobacillus* spp. exhibit antihypertensive and anti-inflammatory properties, making them attractive functional ingredients in nutraceutical formulations [113,114]. Advancements in bioprocess optimization, high-yield proteolytic enzyme engineering, and microbial strain selection have further improved peptide production efficiency, leading to expanded applications in food, pharmaceutical, and cosmeceutical industries.

### 14.3. Challenges and Future Perspectives

Despite the growing industrial significance of bioactive peptides (BAPs), several challenges must be addressed to enhance their scalability, economic feasibility, and commercial viability.

#### 14.3.1. Low Yield and Prolonged Fermentation Time

Variability in microbial proteolytic enzyme activity often leads to low peptide yields and extended fermentation times, posing a challenge to large-scale production. Improving enzyme efficiency and optimizing fermentation parameters are critical for enhancing industrial productivity.

#### 14.3.2. Sensitivity to Environmental Factors

The bioactivity, composition, and stability of BAPs are highly dependent on pH, temperature, and substrate specificity. Variations in fermentation conditions can alter peptide sequences, affecting functional properties and reproducibility and creating consistent large-scale manufacturing challenges.

#### 14.3.3. Challenges in Purification and Downstream Processing

Fermentation-derived peptide mixtures contain bioactive peptides, exopolysaccharides, and microbial debris, making purification costly and technically demanding. While ultrafiltration and high-performance liquid chromatography (HPLC) are widely used, they suffer from membrane fouling, poor scalability, and low reproducibility, limiting industrial applications [114]. To address these challenges, several emerging technologies and biotechnological strategies are being explored:

**Advances in Enzyme and Microbial Engineering:** Innovations in enzyme engineering and metabolic pathway optimization are improving proteolytic cleavage specificity, leading to higher peptide yield and bioactivity. CRISPR-based metabolic engineering enables precise genetic modifications in microbial strains, optimizing pathways to enhance protease efficiency, eliminate bottlenecks, and improve biosynthesis [119].

**Computational Approaches and In Silico Peptide Design:** Computational modeling and in silico peptide prediction allow the rational design of peptides with enhanced bioavailability, stability, and functional properties, accelerating the discovery of novel BAPs for diverse applications.

**Cell-Free Systems and Bioprocess Intensification:** Emerging cell-free enzymatic systems offer a sustainable alternative to traditional fermentation, enabling one-pot enzymatic peptide synthesis while reducing purification costs and improving process efficiency. When combined with precision fermentation and bioprocess intensification strategies, these systems provide scalable and cost-effective solutions for peptide production [120].

With ongoing advancements in biotechnology, synthetic biology, and metabolic engineering, microbial bioactive peptide production is evolving into a scalable, sustainable, and commercially viable alternative to traditional sources. These innovations are expected to accelerate industrial adoption, drive market integration, and expand the role of bioactive peptides in functional foods, pharmaceuticals, and health-related applications [113,114,120].

## 15. Comparative Analysis of Traditional vs. Microbial Production

To assess the advantages and trade-offs of microbial production, Table 4 presents a comparison between traditional extraction and fermentation-based microbial synthesis in terms of yield, purification, sustainability, and cost-effectiveness.

## 16. Relevance and Challenges of Commercial Nutraceutical Production

The commercial production of nutraceuticals is expanding rapidly, driven by increasing consumer demand for functional foods and health-enhancing supplements [117]. However, scaling up microbial bioproduction presents several challenges, particularly in optimizing microbial strains and improving manufacturing efficiency. Microbial strains must be carefully selected and studied to ensure that they can withstand processing conditions, utilize cost-effective substrates such as molasses, whey, starch, and agro-industrial residues, and maintain consistent productivity [18,100]. A well-controlled fermentation process is essential to achieving high yields and product quality. Techniques such as batch, fed-batch, and perfusion fermentation are commonly used, with strict monitoring through Process Analytical Technology (PAT) tools to maintain precision and reproducibility [37,91]. Despite advances, limitations in microbial metabolism and process efficiency still hinder large-scale production.

To overcome these challenges, advanced biotechnological approaches are being explored to enhance strain performance and streamline production. Omics technologies, including genomics, transcriptomics, proteomics, and metabolomics, provide valuable insights into microbial metabolism, enabling the development of more robust strains with improved productivity [18,88,89].

Analyzing host genomics has provided insights into how the host genome shapes microbiome diversity and influences host phenotypes [121]. Likewise, metabolomics sheds light on the metabolic functions of the gut microbiome and its impact on the host gut metabolome [122]. Consequently, a multi-omics approach that integrates metagenomics and untargeted metabolomics at both intra- and inter-domain levels has uncovered interactions between host and microbial metabolites, offering valuable insights into the microbiota’s role in aging [122]. Additionally, genomics plays a key role in identifying probiotic potential within commensal microbes [123], while metagenomics helps decipher the interactions between probiotics and the microbiome [124]. The integration of transcriptomics, proteomics, and metabolomics provides a comprehensive profile of host–microbe interactions and the effects of probiotic supplementation on the host [125]. Transcriptomics, particularly through RNA sequencing or microarray gene expression analysis, is crucial for determining the immunomodulatory properties of probiotics. Meanwhile, proteomics utilizes protein chips containing antibodies, nucleic acids, or other protein-binding molecules to assess proteome changes [126]. The application of multi-omics techniques facilitates the purification of antimicrobial peptides from microbes through liquid chromatography–tandem mass spectrometry (LC–MS/MS), while integrated genomics and proteomics analyses help characterize the genes responsible for these compounds [127]. This approach also supports evolutionary studies through comparative genomics and the discovery of novel antimicrobial compounds. Mass spectrometry plays a critical role in identifying antimicrobial peptide sequences and their molecular masses within complex mixtures, providing insights into their structural and functional properties [128,129,130]. The next section will discuss these innovative strategies and their role in optimizing nutraceutical production.

### 16.1. Strain Selection for Nutraceutical Production

Several studies have focused on improving fermentation techniques, emphasizing enhanced functionality, better hygiene procedures, increased yields, and the standardization of fermentation processes using specific starter strains [5]. Generally Recognized as Safe (GRAS) microorganisms are widely used in industry due to their safety and versatility in pharmaceutical and food applications. Microbial cell factories such as *S. cerevisiae*, *Y. lipolytica*, *A. pullulans*, *Corynebacterium glutamicum*, *E. coli*, *Lactiplantibacillus plantarum*, and *Lactiplantibacillus rhamnosus* offer significant advantages for producing high-value nutraceuticals, including polyphenols, omega-3 fatty acids, vitamins, and probiotics [118,119]. The regulation of genetically modified microorganisms (GMMs) for nutraceutical production presents ethical, legal, and social challenges. Lawmakers, regulatory bodies, and industry stakeholders must address these concerns as biotechnology laws continue to evolve. The challenge lies in balancing technological advancements with safety, fairness, and public confidence [131,132,133]. The World Health Organization (WHO) plays a key role in setting international biotechnology standards. In the United States, multiple federal agencies regulate biotechnology, each with specific responsibilities. The Food and Drug Administration (FDA) oversees the approval of biotechnology products related to human health, such as genetically modified drugs, vaccines, and gene therapies. This process includes extensive pre-market testing, clinical trials, and post-market monitoring to assess potential risks, including allergic reactions, unintended genetic modifications, and long-term health effects [131,132,133].

In Europe, biotechnology regulation is managed at both national and European Union (EU) levels. The European Medicines Agency (EMA) regulates gene therapies, cell-based treatments, and biologics. The EU has some of the world’s strictest regulations for GMMs, and public resistance remains high in several member states. Regulatory approaches to GMMs vary across countries [132,134]. Canada, Australia, and Japan also have their own regulatory frameworks. However, the absence of global agreement on key issues such as GMM oversight, biotechnology patent rights, and ethical concerns related to gene editing makes establishing uniform international standards challenging [132,134].

### 16.2. Synthetic Biology and Heterologous Expression

Synthetic biology serves as a cornerstone of modern nutraceutical production, offering unprecedented control over microbial biosynthetic potential. It has revolutionized the production of complex nutraceuticals by enabling the creation of novel metabolic pathways in microorganisms. One of the most significant advantages of synthetic biology is its ability to combine genes from different organisms, creating biosynthetic pathways that do not naturally exist in a single organism. For example, the heterologous expression of plant genes in yeast as a microbial host has facilitated the yeast synthesis of plant natural products (PNPs) such as artemisinic acid and farnesene [120,135]. Traditionally extracted from plants, these compounds can now be produced at scale using engineered yeasts, eliminating issues like seasonal availability and low yields from natural sources. Furthermore, *E. coli* has been engineered with genes from various plant species to produce PNPs such as resveratrol, a compound known for its anti-inflammatory and antioxidant properties [136,137].

Another notable achievement in synthetic biology is the ability to rewire metabolic fluxes to boost the production of key intermediates. For instance, *S. cerevisiae* has been optimized to increase the production of artemisinic acid, a precursor to the antimalarial drug artemisinin, by redirecting carbon flux toward the mevalonate pathway [138]. Such advances make microbial platforms viable alternatives to plant-based extraction, reducing environmental impact and enhancing sustainability.

Synthetic biology has also enabled the production of a wide range of nutraceuticals. Flavonoids such as naringenin and bioactive peptides have been successfully synthesized in microbial hosts by introducing heterologous biosynthetic pathways [79]. Compared to traditional extraction methods, microbial production offers a more efficient, cost-effective, and scalable process. Furthermore, synthetic biology allows for improvements in the bioavailability and stability of nutraceuticals. Advanced techniques, such as enzyme engineering and pathway optimization, have led to higher yields, improved purity, and enhanced functional properties of the final products, addressing the growing global demand for nutraceuticals [107,139].

Through advanced synthetic biology techniques, researchers can redesign or construct entirely new metabolic networks in microbial hosts like *E. coli* and *S. cerevisiae*, enabling the efficient production of high-value nutraceuticals such as polyphenols, carotenoids, and bioactive peptides. These compounds, often difficult or inefficient to extract from natural sources, can be sustainably and scalably produced through microbial fermentation [140]. An additional advantage of this approach is the use of non-food lignocellulosic feedstocks, such as agricultural residues and municipal waste. These alternative substrates provide an environmentally friendly and cost-effective solution for generating valuable bioactive compounds, reducing reliance on traditional food crops and enhancing scalability [89].

### 16.3. Advances in CRISPR/Cas9 Technology

One of the most transformative technologies in microbial genetic engineering is CRISPR/Cas9, a genome editing tool that allows for precise modifications of microbial DNA. This tool enables scientists to tailor microbial metabolism for enhanced nutraceutical production by knocking out undesirable genes or introducing new biosynthetic pathways, thereby improving the efficiency of bioactive compound production. CRISPR/Cas9 has been developed to enable precise genetic editing, optimizing production pathways and stoichiometry while addressing challenges such as expanding carbon utilization, reducing metabolic burden, and enhancing strain stability. For instance, *S. cerevisiae* has been successfully engineered using CRISPR/Cas9 to boost the production of resveratrol [141].

In addition, CRISPR/Cas9 was successfully employed in a comprehensive metabolic engineering study to expand the carbon fermentation capabilities of *S. cerevisiae* to include glycerol [142]. This strategy demonstrated promising applications for utilizing biomass resources in bioethanol production [143,144]. Moreover, a similar CRISPR/Cas9 approach enabled *S. cerevisiae* to convert glycerol into 2,3-butanediol [145]. These engineered *S. cerevisiae* strains hold significant potential as platforms for redirecting metabolic fluxes toward nutraceutical production, particularly when targeting production from agricultural residues and acid-catalyzed glycerolysis [144].

Notably, *Y. lipolytica* efficiently utilizes glycerol natively for citric acid production [146]. Furthermore, the application of CRISPR/Cas9 genome editing and UV mutagenesis enhanced the conversion of glycerol to erythritol, a nutraceutical compound, achieving production titers as high as 150 g/L, with yields of 0.62 g/g and productivities of 1.25 g/L/h [147]. Ongoing research and development in such recombinant strains, as well as further synthetic pathways or heterologous expression of effective genes, are crucial for overcoming current limitations and achieving commercial viability.

Therefore, the capability of synthetic biology and heterologous expression pathways not only enhances production efficiency but also contributes to the sustainability of nutraceutical production by reducing reliance on traditional food-based feedstocks. However, despite the efficiency gains provided by CRISPR/Cas9, significant challenges remain, particularly in the regulatory landscape governing genetically modified organisms (GMOs). In markets with stringent labeling and approval requirements, such as the European Union, regulatory hurdles can delay or complicate the commercial use of CRISPR-modified organisms. Addressing these regulatory barriers will be critical for expanding the commercial use of CRISPR/Cas9 in nutraceutical production. Effective collaboration with regulatory agencies, transparent safety testing, and consumer education will be essential to overcome these challenges and foster broader acceptance of GMO-derived nutraceuticals.

### 16.4. Adaptive Evolution: Enhancing Microbial Efficiency

Adaptive evolution is an effective strategy in microbial strain development that leverages selective pressure to enhance microorganisms’ nutrient utilization and metabolite production capabilities. Unlike targeted genetic engineering, this method simulates natural selection by subjecting microbial populations to specific environmental stressors, leading to the development of strains with improved traits.

In adaptive evolution, microorganisms are cultivated under controlled conditions where selective pressures, such as nutrient scarcity or exposure to toxic byproducts, promote the survival of mutants with advantageous adaptations. These mutations can result in increased metabolic efficiency, enhanced tolerance to stressful conditions, or elevated production of target compounds like polyphenols, probiotics, and fatty acids. Research has demonstrated the successful application of adaptive evolution to enhance microbial strains for nutraceutical production. For example, *S. cerevisiae* has evolved to tolerate high ethanol concentrations—a toxic byproduct of fermentation—significantly boosting bioethanol production [148]. *E. coli* has been adapted to enhance GABA production using glycerol as the carbon source [69]. A heavy-ion mutagenesis was used in adaptive evolution of *Aurantiochytrium* sp. and increased the production of DHA from 0.18 to 0.27 g/L/h and the titer yield from 21 to 27 g/L [90].

One of the key advantages of adaptive evolution is its potential to enhance microbial performance without extensive genetic modification. This method promotes the natural development of beneficial traits, which can then be integrated with genetic engineering techniques for further optimization. Moreover, adaptive evolution is applicable to a broad range of microorganisms, including those that are challenging to modify using conventional genetic tools. Another benefit is the long-term stability of the evolved traits. Once a microbial strain adapts to a particular environment, its enhanced characteristics are generally preserved over successive generations, making adaptive evolution a robust and dependable method for strain improvement in industrial applications [149].

### 16.5. Fermentation Technologies for Nutraceutical Production

Fermentation remains the backbone of microbial production for nutraceuticals, providing a controlled environment where microorganisms can efficiently produce bioactive compounds such as polyphenols, probiotics, and fatty acids. Bioreactors enable precise control over factors such as temperature, pH, and oxygen levels, optimizing growth conditions and enhancing production efficiency. Advances in fermentation methods have improved yields, reduced costs, and facilitated scalability to meet global demand for nutraceuticals [139,141,150].

Batch fermentation is one of the most traditional and widely used methods for producing nutraceuticals. In this closed system, all ingredients are added at the beginning, and fermentation proceeds without any further input until completion. This method is particularly suitable for the production of probiotics and vitamins, where strict control of microbial growth and nutrient depletion is essential. Lactic acid bacteria (LAB), commonly used in batch fermentation, play a key role in probiotic production [150].

To overcome the limitations of batch fermentation, fed-batch fermentation allows for the continuous addition of nutrients during the fermentation process [150]. This ensures that microorganisms continue to grow and produce metabolites at optimal rates without being inhibited by nutrient depletion or toxic byproducts [151]. Fed-batch fermentation is commonly used to produce carotenoids such as β-carotene from *Y. lipolytica* [152], as well as from *S. cerevisiae* [153]. The production rate and total yield of β-carotene are significantly different when using bioreactors compared to flasks. The *S. cerevisiae* SM14 strain, developed through adaptive evolution, produced up to 21 mg/g dry cell weight (DCW) of β-carotene in shake flask cultures, while the βcar1.2 strain, generated by overexpressing carotenogenic genes, produced only 5.8 mg/g DCW. In fed-batch bioreactors, however, βcar1.2 outperformed SM14, achieving higher biomass and β-carotene productivity rates of 1.57 g/L/h and 10.9 mg/L/h, respectively, compared to SM14’s 0.48 g/L/h and 3.1 mg/L/h [154]. By continuous glucose feeding, the yeast cells maintain optimal metabolic activity, leading to 10.5% higher β-carotene production [153]. Continuous fermentation systems allow for the steady addition of nutrients and the removal of products and byproducts in real time, creating an uninterrupted production cycle. Microorganisms are kept in their exponential growth phase, where they are most productive, and products are harvested continuously. Continuous fermentation has been successfully employed in the production of Bifidobacterium species, a probiotic used in supplements to improve gut health. The system allows for the constant production of probiotic cultures, ensuring consistent quality and quantity of the product [155].

Solid-state fermentation involves growing microorganisms on solid substrates rather than in liquid media. This method is gaining attention for producing nutraceuticals with unique properties. *Aspergillus oryzae* is used in solid-state fermentation to produce antioxidants from agricultural byproducts, such as rice bran, for use in nutraceuticals [156]. Co-cultivation involves growing multiple microbial strains in the same bioreactor to exploit their synergistic interactions. Co-cultivation of Lactobacillus and Bifidobacterium species in the same bioreactor has been used to enhance the production of multi-strain probiotic supplements. These probiotics work together to improve the balance of gut microflora [157]. Membrane bioreactors (MBRs) use semi-permeable membranes to separate biomass from the fermentation broth, allowing for continuous filtration and product extraction. Membrane bioreactors are employed in the production of amino acids such as L-lysine by *C. glutamicum*. The MBR system allows for continuous removal of L-lysine, enhancing overall yield and simplifying downstream processing [141].

### 16.6. Process Optimization

The efficiency and yield of microbial fermentation for nutraceutical production can be significantly improved through process optimization. Critical parameters such as pH, temperature, oxygen levels, and nutrient supply must be precisely controlled to ensure the optimal growth of microorganisms and the production of desired metabolites [158]. Modern fermentation processes now integrate advanced monitoring and control systems, such as Process Analytical Technology (PAT), which provides real-time monitoring of key fermentation parameters, allowing for dynamic adjustments to optimize the fermentation process [159]. In the production of resveratrol by engineered *E. coli*, PAT tools are used to monitor the concentration of glucose, oxygen levels, and biomass during fermentation. These real-time data allow for fine-tuning of the process to maximize carotenoid yield [159]. In the production of omega-3 fatty acids from Schizochytrium species, a marine microalga, maintaining precise control over oxygen levels and pH, is critical to optimizing lipid accumulation, which is essential for high yields of omega-3 fatty acids. Automated bioreactors are employed to continuously monitor and adjust these parameters, ensuring that the microalga remains in its optimal growth phase throughout the fermentation process [160].

Higher yields and quality, along with lower costs and waste generation, will result from being able to depend on automated smart systems that need minimal human/manual intervention, which is crucial for bio-based biopharma products, particularly nutraceuticals. Gathering extensive sets of pertinent data is the greatest barrier to the adoption of smart manufacturing in bio-based industries [161,162]. According to studies, creating mathematical models has many advantages, particularly for the control and optimization of bioprocesses, to ensure operational reproducibility, quality control, and consistency [162]. As a result, sensors are needed to monitor physical factors like temperature and pressure, chemical quantities like pH and dissolved oxygen, and biological characteristics like cell density or metabolite concentrations. As shown in Figure 6, monitoring techniques and the sensors and analyzers that go along with them can be further categorized based on where they are in relation to the process unit. An in-line sensor constantly generates data (no sampling), and it is either in direct contact with the process medium (invasive) or is separated from it by a glass window (also known as an on-line sensor, non-invasive). These sensors enable continuous process control by providing continuous information. Samples close to the bioreactor are analyzed by at-line sensors. Even though the samples are collected regularly (manually or automatically), the analysis-related time delays (depending on the equipment) make such data ideal for monitoring but not for control. Finally, off-line measurement samples are manually or automatically collected before being sent to the lab for analysis. Due to the lengthy delays that result, these measurements are unable to control the dynamic process behavior [163].

## 17. Microbial Intelligence (MI)

Microbial intelligence encompasses microorganisms’ complex behaviors and decision-making processes. These processes allow them to adapt to environmental changes, communicate, and form intricate community structures. These intelligence strategies enable microorganisms to thrive in diverse environments and are increasingly leveraged in producing nutraceuticals. Microbial communication, as one of these intelligent mechanisms, is integral to the production of nutraceuticals. Quorum sensing (QS) regulates gene expression related to secondary metabolite production, impacting nutraceutical synthesis [164]. QS can be auto-induced at the bacterial population level by molecules such as acyl-homoserine lactones, peptides (AIPs), and indole, and highlight the potential in developing novel clinical therapies [165]. Complex signal processing in synthetic gene circuits using cooperative regulatory assemblies was elegantly studied. It showed the application of cooperative regulatory assemblies in synthetic biology to program nonlinear gene circuit behaviors. It also highlights how these systems can expand engineerable signal-processing capabilities in synthetic networks, with potential applications in various biological and technological contexts [166]. This QS is attractive more broadly, with studies focusing on probiotics, where cross-talks between different genera of fungi have been frequently observed in co-culturing, giving rise to altered profiles of secondary metabolites [167]. The potential of epoxide-containing compounds, such as cerulenin and fosfomycin, as quorum-sensing inhibitors (*QSIs*) for *Staphylococcus aureus* and *Enterococcus faecalis* was studied. These compounds were tested for their inhibitory effects on quorum-sensing systems (*agr* and *fsr*) at sublethal concentrations. The results showed that both compounds significantly inhibited gelatinase production mediated by the *fsr* system in *E. faecalis* and reduced GFP and luciferase expression in *S. aureus*. Molecular docking revealed that their epoxide groups interacted with active sites in key quorum-sensing proteins, demonstrating their potential as antimicrobial agents targeting quorum-sensing pathways [168]. This reduces the likelihood of resistance development, which is a key challenge in probiotic therapy aimed at balancing the microbiome.

Microbial intelligence (MI) plays a crucial role in probiotic function and nutraceutical production. In *Bifidobacteria*, quorum sensing (QS) regulates the secretion of bioactive metabolites, including extracellular vesicles and biofilms. The QS system, particularly the *luxS/AI-2* pathway, also inhibits virulence gene expression in opportunistic microorganisms. Additionally, QS enhances host–microbe interactions, contributing to improved gut health and better nutritional status [169]. Another example of MI in nutraceutical production is the biosynthesis of conjugated linoleic acid (CLA) by *Limosilactobacillus fermentum* L1. Metabolomic and biochemical studies reveal that QS directly regulates high CLA production. Inhibition of QS significantly reduces CLA yield, with production dropping by nearly half when quorum-sensing inhibitors are applied (*p* < 0.01). Metabolomic analysis identified 306 differential metabolites between high- and low-CLA-producing strains, showing that QS influences key metabolic pathways such as energy metabolism, signal transduction, and redox balance. Notably, glutathione and malondialdehyde levels increased in high-CLA producers, indicating that QS-mediated redox regulation enhances CLA synthesis [170]. In *Weissella confusa* XG-3, low exopolysaccharide (EPS) yield is a limiting factor. However, co-cultivation with *Candida shehatae* enhances EPS production. Structural analysis of purified EPS reveals that *C. shehatae* promotes *W. confusa* XG-3 growth by consuming organic acids in the culture medium, triggering QS, upregulating *luxS*, *ackA*, and *wzb* gene expression, and increasing dextransucrase activity. This interaction stimulates dextran biosynthesis, demonstrating how microbial co-cultivation can optimize metabolite production [171].

### Unlocking the Potential of Microbial Nutraceuticals

Standardization and quality control remain critical, as variability in microbial strains, fermentation processes, and product formulations can affect the quality, efficacy, and safety of these products. Ensuring consistency requires strict quality control measures and a deep understanding of fermentation conditions. Moreover, shelf stability can be a limiting factor, as products containing live microorganisms may require refrigeration, impacting their accessibility and convenience [172].

Another challenge lies in the efficacy of microbial supplements, which can vary significantly among individuals. Factors such as gut microbiota composition, overall health status, and individual biological responses influence the effectiveness of these products. Regulatory considerations further complicate the landscape, as compliance with diverse and region-specific standards creates uncertainties for manufacturers, particularly in the absence of globally standardized guidelines [173,174].

Finally, a thorough understanding and precise control of fermentation processes are crucial for consistently producing high-quality nutraceuticals. Continuous research into the mechanisms of action, optimal dosages, and potential interactions with medications is pivotal to unlocking the full potential of microbial nutraceuticals. By addressing these challenges, microbial nutraceuticals are poised to become a cornerstone of the functional food and dietary supplement industries, aligning with the growing consumer demand for sustainable, health-promoting products.

## Figures and Tables

**Figure 1 microorganisms-13-00566-f001:**
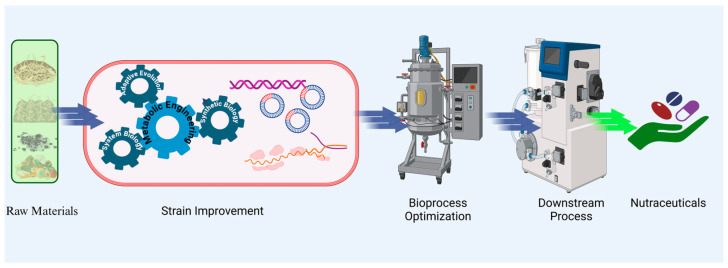
Microbial bioprocessing for nutraceutical production: from raw materials to high-value nutraceutical products. Created with BioRender.com.

**Figure 2 microorganisms-13-00566-f002:**
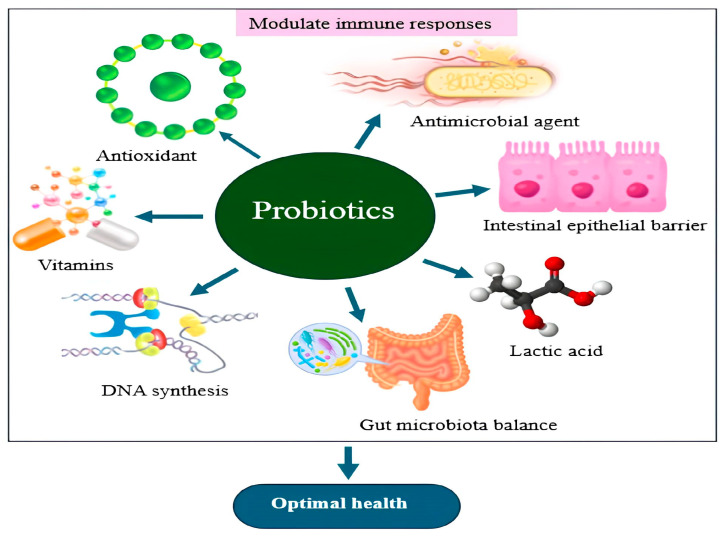
Multifunctional effects of probiotics on health and immune response.

**Figure 3 microorganisms-13-00566-f003:**
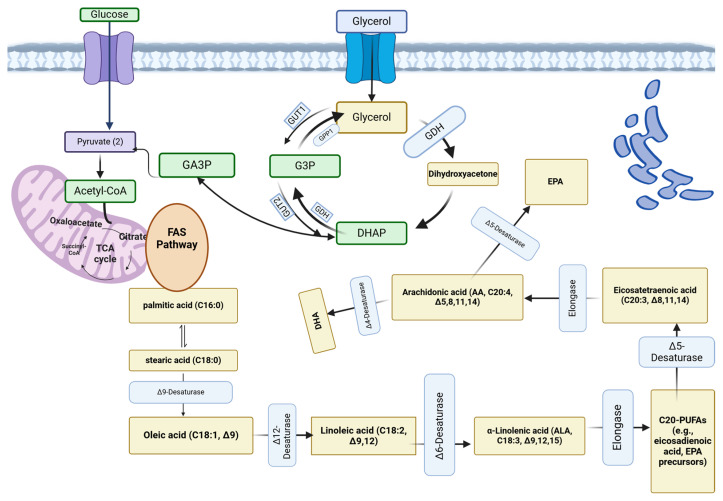
Key steps in PUFA biosynthesis pathway. Created with BioRender.com.

**Figure 4 microorganisms-13-00566-f004:**
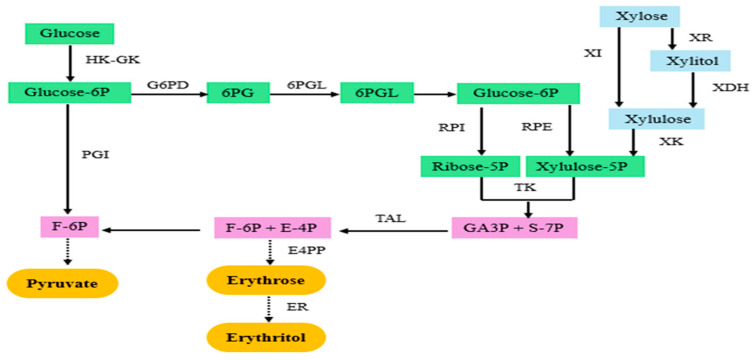
Key enzymatic steps in the erythritol biosynthetic pathway.

**Figure 5 microorganisms-13-00566-f005:**
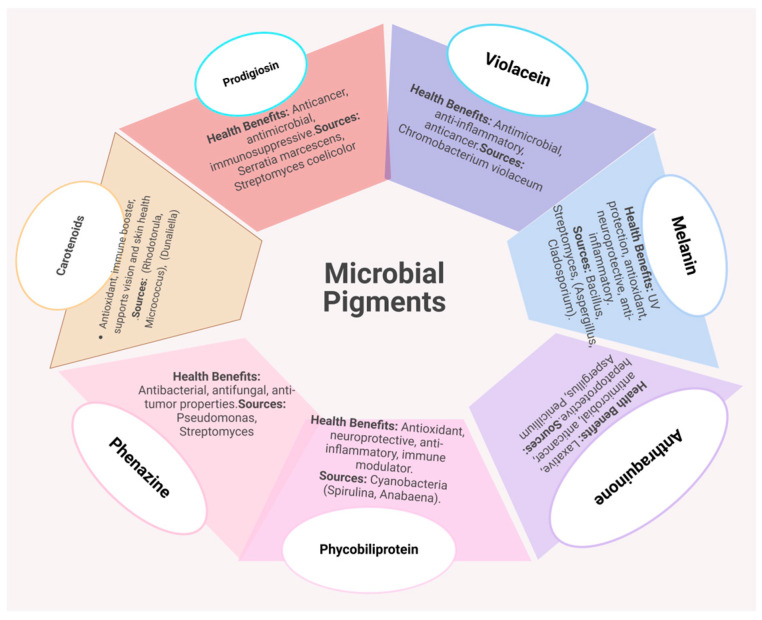
Microbial pigments in nutraceuticals and their health benefits. Created with BioRender.com.

**Figure 6 microorganisms-13-00566-f006:**
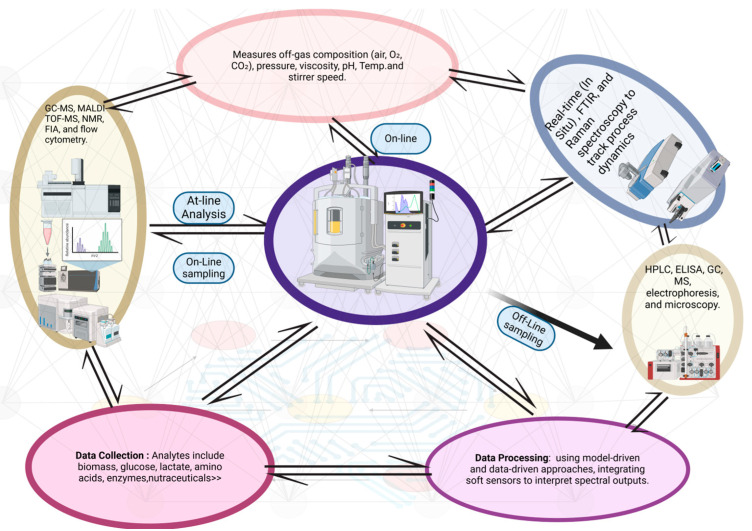
Integrated analytical approaches for real-time and off-line bioprocess monitoring. Created with BioRender.com.

**Table 1 microorganisms-13-00566-t001:** The most prominent and promising oleaginous yeast species and their key characteristics.

Microorganism Type	Species	Biomass (g/L)	Lipid Content (%)	Lipid Titer (g/L)	Overall Yield (g/g)	Lipid Productivity (g/L/h)	Feedstock/Substrate	Lipid Composition Majority	Culture Mode	Reference
Yeast	*Yarrowia lipolytica* NS432	115.75	73	84.5	0.2	0.73	Glucose	(C16:0), (C16:1), (C18:0), (C18:1), and (C18:2)	Fed-batch bioreactor	[39]
		40	77	30.8	0.21	0.27			Batch bioreactor	
	*Yarrowia lipolytica*	89.3	81.4	72.7	0.252	0.97	Glucose	(C18:1) 50% and (C18:2) 10%	Stepwise exponential feeding bioreactor	[40]
	*Cryptococcus curvatus*	4.32	41.2	1.78	0.11	0.037	Waste office paper	(C18:1) almost around 50%, (C16:0), (C18:0), and (C18:2)	Batch shake flask	[41]
	*Lipomyces starkeyi*	85.4	49	41.8	0.1	0.176	Xylose, glucose	(C14:0), (C16:0), (C18:0), (C18:1), (C18:2), and (C18:3)	Repeated fed-batch bioreactor	[42]
	*Lipomyces starkeyi*	13.30	60.47	8.04	0.11	0.0558	Hemicellulose hydrolysate	(C16:0), (C18:1), (C18:0), (C18:2), and (C18:3)	pH-regulated fed-batch cultivation, bioreactor	[43]
	*Rhodosporidium toruloides*	26.8	61.1	16.4	0.23	0.08	Glucose	(C18:1) and (C16:0)	Batch shake flask	[44]
	*Rhodotorula glutinis*	70.8	47.2	33.5	0.159	0.17	Undetoxified corncob hydrolysate	(C16:0), (C18:1), and (C18:2)	Fed-batch culture with two-stage nitrogen feeding strategy in bioreactor	[45]
	*Candida tropicalis ASY2*	2.49	48.59	1.21	0.135	0.01	Starch Wastewater (SWW)	(C18:1) and (C20:1)	Batch-flask	[46]
	*Candida viswanathii Y-E4*	26.2	51.9	13.6	13.6	0.08	Crude glycerol	(C18:1) and (C18:2)	Batch-flask	[47]
	*Pichia kudriavzevii*	33.2	57.9	19.2	0.162	0.296	Acetic and propionic acid	(C16:0), (C18:0), (C18:2), and (C17:0)	Fed-batch bioreactor	[48]

**Table 2 microorganisms-13-00566-t002:** Comparative analysis of microbial exopolysaccharides (EPSs) including microbial sources, key biosynthetic enzymes, industrial applications, and reported yields. This table highlights enzymatic diversity, production trade-offs, and commercial relevance.

Exopolysaccharide	Microbial Source	Key Biosynthetic Enzymes	Industrial Applications	Yield (g/L)	Bioactivity and Mechanism	Reference
Xanthan	*Xanthomonas campestris*	UDP-glucose pyrophosphorylase, glycosyltransferases	Food, cosmetics, biomedicine	50+	Antioxidant (ROS scavenging, Nrf2 activation)	[50]
Hyaluronic Acid	*Streptococcus zooepidemicus*	Hyaluronan synthase	Pharmaceuticals, skincare, tissue engineering	15	Immunomodulation (TLR2, cytokine activation)	[56]
Alginate	*Pseudomonas aeruginosa*	Alginate polymerase	Food, wound healing, biofilms	30	Prebiotic (SCFA production, gut microbiota)	[51]
Levan	*Bacillus subtilis*	Levansucrase	Prebiotics, drug delivery	40	Antimicrobial (biofilm inhibition)	[50]
Scleroglucan	*Sclerotium rolfsii*	Beta-glucan synthases	Pharmaceuticals, antiviral agents	66.6	Immunostimulatory (TLR4 activation, TNF-α secretion)	[55]

**Table 3 microorganisms-13-00566-t003:** Comparative analysis of bioactive peptides.

Bioactive Peptide	Microbial Source	Bioactive Mechanism	Industrial Applications	Reference
Lactotripeptides (VPP, IPP)	*Lactobacillus helveticus*	ACE inhibition (antihypertensive)	Functional dairy, cardiovascular health	[115]
Casein-derived peptides	*Lactobacillus casei*	Antioxidant (ROS scavenging, Nrf2 activation)	Nutraceuticals, anti-aging formulations	[116]
Soy-derived peptides	*Bifidobacterium longum*	Immunomodulation (TLR2-mediated cytokine activation)	Functional foods, gut health	[117]
Antimicrobial peptides (AMPs)	*Bacillus subtilis*	Membrane disruption, quorum sensing inhibition	Food preservatives, antibiotics	[118]
Lactotripeptides (VPP, IPP)	*Lactobacillus helveticus*	ACE inhibition (antihypertensive)	Functional dairy, cardiovascular health	[115]
Casein-derived peptides	*Lactobacillus casei*	Antioxidant (ROS scavenging, Nrf2 activation)	Nutraceuticals, anti-aging formulations	[116]
Soy-derived peptides	*Bifidobacterium longum*	Immunomodulation (TLR2-mediated cytokine activation)	Functional foods, gut health	[117]
Antimicrobial peptides (AMPs)	*Bacillus subtilis*	Membrane disruption, quorum sensing inhibition	Food preservatives, antibiotics	[118]

**Table 4 microorganisms-13-00566-t004:** Comparative analysis of traditional vs. microbial production: trade-offs, cost, and sustainability.

Criteria	Traditional Production (Plant/Animal-Based)	Microbial Production (Fermentation-Based)	Effectiveness (Microbial vs. Traditional)
Source	Extracted from plants or animal tissues (e.g., bovine cartilage, fish collagen, plant-derived polysaccharides, chemically synthesized lactic acid)	Produced via microbial fermentation (*Lactobacillus*, *Bacillus*, *Streptococcus*, *Xanthomonas*, *Sclerotium rolfsii*)	Microbial production is more sustainable and scalable, avoiding seasonal/raw material fluctuations
Yield	Variable, seasonal limitations (e.g., 0.2–2 g/L for plant-extracted polysaccharides, 0.5–1 g/L for lactic acid from petrochemical sources)	High, stable, and scalable (e.g., 50–80 g/L for microbial EPS, 10–20 g/L for bioactive peptides, 66.6 g/L for scleroglucan, 100–150 g/L for lactic acid fermentation)	Microbial yields are significantly higher (10–50× improvement), making production more efficient
Purification Process	High cost, low efficiency—contamination risk, batch-to-batch variation, difficult to remove unwanted plant/animal residues, high refining cost for petrochemical lactic acid	Lower cost, higher efficiency—controlled microbial expression reduces unwanted byproducts, enabling cost-effective downstream processing	Microbial production is more cost-effective due to controlled biosynthesis, reducing unwanted impurities
Purification Cost and Challenges	High cost, low efficiency—contamination risk, batch-to-batch variation, difficult to remove unwanted plant/animal residues, high refining cost for petrochemical lactic acid	Lower cost, higher efficiency—controlled microbial expression reduces unwanted byproducts, enabling cost-effective downstream processing	Microbial production is more cost-effective due to controlled biosynthesis, reducing unwanted impurities
Bioactivity Retention	Can degrade due to processing conditions (e.g., heat, solvent exposure)	Controlled fermentation preserves bioactivity, optimizing peptide/enzyme activity	Microbial processes offer higher bioactivity retention
Purity and Consistency	Variable purity—batch-to-batch variation, presence of contaminants (e.g., heavy metals, allergens, chemical residues in lactic acid synthesis)	Higher purity and consistency—controlled biosynthesis minimizes contaminants	Microbial systems provide higher reproducibility and purity
Sustainability	High land, water, and energy consumption, reliance on petrochemical sources (lactic acid)	Lower environmental footprint, sustainable production	Microbial processes are greener, reducing resource use
Cost	Dependent on agricultural fluctuations, higher raw material costs, chemical synthesis cost for lactic acid	Lower long-term cost after process optimization	Microbial production is more cost-effective long-term
Industrial Examples	- Collagen from bovine sources (3–5% extraction efficiency) - Plant-derived flavonoids - Soy peptides - Lactic acid from petrochemical sources (yield ~0.5–1 g/L)	- Hyaluronic acid from *Streptococcus zooepidemicus* (~15 g/L) - Bioactive peptides from *Lactobacillus* (10–20 g/L) - Xanthan gum from *Xanthomonas campestris* (~50 g/L) - Scleroglucan from *Sclerotium rolfsii* (66.6 g/L) - Lactic acid from *Lactobacillus* spp. and *Bacillus* spp. (100–150 g/L)	Microbial production dominates high-value functional ingredients, especially for scleroglucan, lactic acid, xanthan, and bioactive peptides

## Data Availability

No new data were created or analyzed in this study.

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
