# Peer review of "Where Biology Meets Engineering: Scaling Up Microbial Nutraceuticals to Bridge Nutrition, Therapeutics, and Global Impact"

_microorganisms, 2025, doi:10.3390/microorganisms13030566_

Round 1

Reviewer 1 Report

Comments and Suggestions for Authors

The Review is very interesting and of high relevance, exploring the current challenges in microbial nutraceutical production. The article contains a lot of relevant information and shows good documentation.

Abstract.

Lines 31-33 check and complete the end of the sentence.

Relevant keywords.

Line 387 please delete repeated words.

Lines 409-411 please be more specific. “Using microorganisms, polyphenols can be converted into metabolites with improved solubility, stability, and bioactivity [77]. Polyphenols, as prominent bioactive compounds in edible fungi,..” please mention the species implied in the bioconversion of phenolic compounds (PC), and respectively fungi that contain PC.

Lines 421-424 please add the species and correct the writing.

Lines 499-508 Carotenoids. Please add phew words about Rhodotorula spp.

Lines 632-634 reference needed. Please add.

Line 691 please add reference number instead of (Fenster et al., 2019).

Figure 8 is not cited in text. Can be removed (“fermentation optimization” is repeated three times). is not relevant to the article.

Lines 694-707 “3. Nutraceuticals Cutting-Edge Technologies” it doesn't say anything concrete. New Cutting-Edge technologies need to be added here. Please add text or remove subsection.

Lines 758-762 a reference is necessary.

Line 832 “Probiotics and vitamins, where strict control of microbial growth and nutrient depletion is necessary.” Please rephrase.

The structure of the article is not appropriate (the main sections are missing). There is an Abstract and an Introduction, but doesn’t have the M&M and Conclusions clearly outlined. Unless otherwise stated, please separate the article into the main sections.

All References are included, recent and relevant to the topic discussed.

Author Response

"Please see the attachment."
Response to Reviewer 1 Comments:

Dear Reviewer,

We sincerely appreciate your time and effort in reviewing our manuscript. Your constructive feedback has been invaluable in improving the quality and clarity of our work. Below, we provide detailed responses to each of your comments, along with the corresponding revisions in the manuscript. All modifications have been highlighted in the resubmitted document.

Point-by-Point Responses

Comment 1: The review is very interesting and highly relevant, exploring the current challenges in microbial nutraceutical production. The article contains a lot of relevant information and shows good documentation.

Response 1: Thank you for your positive feedback. We appreciate your recognition of the relevance and quality of our review.

Comment 2: Lines 31-33: Please check and complete the end of the sentence.

Response 2: We agree and have corrected the sentence by changing "functional bioactive prod." to "functional bioactive products."

Comment 3: Relevant keywords need to be revised.

Response 3: We have updated the keywords as follows:
From: Microbial Biotechnology, Nutraceuticals, Sustainable Production, 35, Probiotics, Prebiotics, Postbiotics
To: Microbial Biotechnology, Nutraceuticals, Sustainable Production, Probiotics, Prebiotics, Postbiotics

Comment 4: Line 387: Please delete repeated words.

Response 4: The repeated words have been removed.

Comment 5: Lines 409-411: Please specify the species involved in the bioconversion of phenolic compounds (PC) and the fungi that contain PC.

Response 5: We have revised this section to include specific microbial species involved in polyphenol bioconversion and the fungi rich in polyphenols. The revised section now states:

"Microbial biotransformation has emerged as a promising approach to enhance the bioavailability of polyphenols. Using microorganisms such as Lactobacillus plantarum, Lactobacillus brevis, Bacillus spp., Saccharomyces cerevisiae, and Aspergillus niger, polyphenols can be converted into metabolites with improved solubility, stability, and bioactivity. Polyphenols, as prominent bioactive compounds in edible fungi such as Lentinula edodes (shiitake), Pleurotus ostreatus (oyster mushroom), Agaricus bisporus (white button mushroom), and Ganoderma lucidum (reishi mushroom), have garnered attention for their significant nutritional and medicinal value. Notable fungal-derived polyphenols, such as quercetin and kaempferol, exhibit potent antioxidant, anti-inflammatory, and anticarcinogenic properties."

Comment 6: Lines 421-424: Please add the species and correct the writing.

Response 6: We have modified the text as follows:

"Microbial production of vitamins has emerged as a superior alternative to traditional chemical synthesis due to its flexibility in microorganism selection, the use of renewable biomass, and environmentally sustainable processes. Various microorganisms, including Bacillus subtilis, Propionibacterium freudenreichii, Saccharomyces cerevisiae, and Rhizopus oryzae, are widely utilized for the biosynthesis of essential vitamins such as riboflavin (B2), cobalamin (B12), folate (B9), and biotin (B7). These microbial processes offer high yield, cost-effectiveness, and improved bioavailability, making them a preferred method for vitamin production in the nutraceutical and pharmaceutical industries."

Comment 7: Lines 499-508: Please add a few words about Rhodotorula spp.

Response 7: We have revised the section to include Rhodotorula spp.:

"Carotenoids are vital bioactive compounds with antioxidant, anti-inflammatory, and pro-vitamin A properties, making them valuable for nutraceutical, pharmaceutical, food, and cosmetic industries. Although synthetic carotenoids dominate the market, concerns over chemical waste and health risks have increased interest in microbial production as a sustainable alternative. Microorganisms such as fungi (Blakeslea trispora, Phycomyces blakesleeanus), yeast (Rhodotorula spp., Xanthophyllomyces dendrorhous), bacteria (Corynebacterium michiganense, Dietzia natronolimnaea), and microalgae (Haematococcus pluvialis, Dunaliella salina) efficiently produce carotenoids under controlled conditions. Rhodotorula spp., widely studied for carotenoid biosynthesis, produces β-carotene, torulene, and torularhodin, with yields influenced by environmental conditions. Microbial fermentation offers advantages such as cost-effective production using low-cost substrates, season-independent processing, and high efficiency, positioning it as a promising alternative to synthetic carotenoids."

Comment 8: Lines 632-634: Reference needed.

Response 8: We have added the appropriate reference.

Comment 9: Line 691: Please replace "(Fenster et al., 2019)" with the corresponding reference number.

Response 9: We have revised the citation and replaced it with the correct reference number.

Comment 10: Figure 8 is not cited in the text. It can be removed, as "fermentation optimization" is repeated three times and is not relevant to the article.

Response 10: We agree and have removed Figure 8.

Comment 11: Lines 694-707: The subsection "3. Nutraceuticals Cutting-Edge Technologies" does not provide concrete information. New cutting-edge technologies should be added, or the section should be removed.

Response 11: We have removed the section.

Comment 12: Lines 758-762: A reference is necessary.

Response 12: A reference has been added.

Comment 13: Line 832: "Probiotics and vitamins, where strict control of microbial growth and nutrient depletion is necessary." Please rephrase.

Response 13: We have reworded the sentence for clarity:

"Batch fermentation is one of the most traditional and widely used methods for producing nutraceuticals. In this closed system, all ingredients are added at the beginning, and fermentation proceeds without any further input until completion. This method is particularly suitable for the production of probiotics and vitamins, where strict control of microbial growth and nutrient depletion is essential. Lactic acid bacteria (LAB), commonly used in batch fermentation, play a key role in probiotic production."

Comment 14: The structure of the article is not appropriate (the main sections are missing). While the article has an Abstract and an Introduction, it lacks clearly defined M&M and Conclusion sections. Please separate the article into the main sections.

Response 14: We have structured the article according to MDPI guidelines for review articles.

Final Remark

We appreciate the reviewer's valuable comments and have made the necessary revisions to improve the manuscript. We hope these modifications address your concerns and enhance the clarity and scientific value of our work.

Thank you again for your time and constructive feedback.

Sincerely,

Reviewer 2 Report

Comments and Suggestions for Authors

While covering many nutraceuticals, some sections (e.g., exopolysaccharides, bioactive peptides) might benefit from deeper mechanistic insights.

Please provide specific details on metabolic pathways, yield-improvement strategies, or industrial examples would strengthen the practical dimension.

I believe the manuscript would benefit from a more consolidated regulatory framework, describing international guidelines (e.g., EFSA, FDA). I believei this is especiallyrelevant for genetically modified strains.

I would like to read more about the concept of microbial intelligence and its application to nutraceutical production. Please provide examples or case studies.

Engineering is about trade-offs, cost reduction and sustainability, I believe a table comparing traditional production vs microbial production would benefit your work.

Please elaborate giving ecxamples  on how multi-omics advanced strain improvements. Maybe a short case study.  

Please spell out acronyms like GRAS, LA, DHA, etc., at first occurrence in each major section for clarity.

Congratulations to the authors for compiling an extensive and timely review that underscores the vital intersection of biology and engineering in meeting global health and sustainability needs. After addressing the suggested enhancements, this manuscript will be a strong reference for the scientific community and industry stakeholders exploring microbial-based nutraceuticals.

Author Response

"Please see the attachment." 
Response to Reviewer #2 Comments

Dear Reviewer,

We sincerely appreciate your valuable time and thoughtful comments, which have helped improve the depth and clarity of our manuscript. Below, we provide detailed responses to each of your suggestions and outline the corresponding revisions made. All changes have been incorporated into the manuscript and highlighted accordingly.

  1. Comment: "While covering many nutraceuticals, some sections (e.g., exopolysaccharides, bioactive peptides) might benefit from deeper mechanistic insights."

Response:
Thank you for your valuable suggestion. We have expanded the sections on exopolysaccharides and bioactive peptides, adding deeper mechanistic insights into their biosynthetic pathways, structure-function relationships, and health benefits. These revisions are highlighted in yellow in the respective sections and further summarized in Tables [2] and [3].

  1. Comment: "Please provide specific details on metabolic pathways, yield-improvement strategies, or industrial examples to strengthen the practical dimension."

Response:
We appreciate this recommendation. To enhance the practical relevance of our review, we have:
✔ Expanded our figures and discussion to provide specific details on key metabolic pathways involved in nutraceutical production.
✔ Included strategies for yield improvement, such as metabolic engineering and process optimization.
✔ Added industrial case studies to illustrate real-world applications of these techniques.

  1. Comment: "I believe the manuscript would benefit from a more consolidated regulatory framework, describing international guidelines (e.g., EFSA, FDA). I believe this is especially relevant for genetically modified strains."

Response:
Thank you for this important comment. We have added a detailed discussion on the regulation of genetically modified microorganisms in both the U.S. (FDA) and European countries (EFSA), including their respective approval processes. The revised content can be found on Line 813-820.

  1. Comment: "I would like to read more about the concept of microbial intelligence and its application to nutraceutical production. Please provide examples or case studies."

Response:
We appreciate your interest in microbial intelligence (MI) and have incorporated three case studies demonstrating its impact:

  1. MI in Bifidobacteria as a probiotic, promoting host-microbial interactions.
  2. The role of MI in enhancing conjugated linoleic acid (CLA) production.
  3. How MI improves exopolysaccharide yield in Weissella sp.

These additions can be found on Line 1021-169.

  1. Comment: "Engineering is about trade-offs, cost reduction, and sustainability. I believe a table comparing traditional production vs. microbial production would benefit your work."

Response:
Thank you for this insightful suggestion. We have added Table [4], which compares traditional extraction methods (e.g., plant-based, chemical synthesis) with microbial production in terms of:
✔ Cost
✔ Sustainability
✔ Efficiency
✔ Product consistency

This table highlights the advantages and trade-offs of microbial-based approaches.

  1. Comment: "Please elaborate with examples on how multi-omics has advanced strain improvements. Maybe a short case study."

Response:
We appreciate this directive on the multi-omics approach. To strengthen this section, we have added a case study on how multi-omics techniques influence the gut microbiome and provide insights for discovering novel antimicrobial peptide sequences through metabolomics studies. This addition is on Line 776-799.

  1. Comment:"Please spell out acronyms like GRAS, LA, DHA, etc., at first occurrence in each major section for clarity."

Response:
Thank you for pointing this out. We have ensured that all acronyms (e.g., GRAS, LA, DHA) are spelled out upon their first occurrence in each major section for clarity.

Final Remark

We sincerely appreciate the reviewer's positive feedback and constructive suggestions, which have significantly enhanced the clarity, depth, and impact of our review. Thank you again for your time and thoughtful input.

Sincerely,

 Correponding Authors
